# Auto-regulation of Rab5 GEF activity in Rabex5 by allosteric structural changes, catalytic core dynamics and ubiquitin binding

Janelle Lauer[1], Sandra Segeletz[1], Alice Cezanne[1], Giambattista Guaitoli[2], Francesco Raimondi[3,4], Marc Gentzel[5], Vikram Alva[6], Michael Habeck[7,8], Yannis Kalaidzidis[1], Marius Ueffing[9], Andrei N Lupas[6], Christian Johannes Gloeckner[2,9], Marino Zerial[1]*

[1]Max Planck Institute of Molecular Cell Biology and Genetics, Dresden, Germany; [2]German Center for Neurodegenerative Diseases, Tübingen, Germany; [3]Bioquant, Heidelberg University, Heidelberg, Germany; [4]Heidelberg University Biochemistry Centre (BZH), Heidelberg, Germany; [5]Molecular Analysis-Mass Spectrometry Center for Molecular and Cellular Bioengineering, Technical University Dresden, Dresden, Germany; [6]Max Planck Institute for Developmental Biology, Tuebingen, Germany; [7]Statistical Inverse Problems in Biophysics, Max Planck Institute for Biophysical Chemistry, Göttingen, Germany; [8]Felix Bernstein Institute for Mathematical Statistics in the Biosciences, University of Göttingen, Göttingen, Germany; [9]Center for Ophthalmology, Institute for Ophthalmic Research, University of Tübingen, Tübingen, Germany

**\*For correspondence:**
zerial@mpi-cbg.de

**Competing interests:** The authors declare that no competing interests exist.

**Abstract** Intracellular trafficking depends on the function of Rab GTPases, whose activation is regulated by guanine exchange factors (GEFs). The Rab5 GEF, Rabex5, was previously proposed to be auto-inhibited by its C-terminus. Here, we studied full-length Rabex5 and Rabaptin5 proteins as well as domain deletion Rabex5 mutants using hydrogen deuterium exchange mass spectrometry. We generated a structural model of Rabex5, using chemical cross-linking mass spectrometry and integrative modeling techniques. By correlating structural changes with nucleotide exchange activity for each construct, we uncovered new auto-regulatory roles for the ubiquitin binding domains and the Linker connecting those domains to the catalytic core of Rabex5. We further provide evidence that enhanced dynamics in the catalytic core are linked to catalysis. Our results suggest a more complex auto-regulation mechanism than previously thought and imply that ubiquitin binding serves not only to position Rabex5 but to also control its Rab5 GEF activity through allosteric structural alterations.

## Introduction

Small GTPases were identified almost 40 years ago and the superfamily has grown to include more than 70 human members (*Cherfils and Zeghouf, 2013*; *Rojas et al., 2012*; *Shih et al., 1980*; *Wittinghofer and Vetter, 2011*). These proteins regulate an array of activities such as cell growth and differentiation, organelle biogenesis, intracellular transport, cytoskeletal organization, and cell division (*Cherfils and Zeghouf, 2013*). Activation and deactivation of small GTPases are controlled by cycling through inactive GDP-bound and active GTP-bound states. These cycles are regulated by guanine nucleotide exchange factors (GEFs), and GTPase activating proteins (GAPs) (*Bos et al.,*

*2007*; *Zerial and McBride, 2001*). In addition, most small GTPases carry a C-terminal lipid modification and variable C-terminal amino acid sequences. This provides a means of membrane association and additional layers of control such as extraction and insertion into specific membranes. Following insertion into the proper membrane and GDP/GTP exchange by an appropriate GEF, the active GTPase can associate with effector molecules mediating biological activity and protecting it from membrane extraction by guanine nucleotide dissociation inhibitors (GDIs) (*Cherfils and Zeghouf, 2011*). The activity of GEFs is therefore of primary importance for the regulation of localization and downstream function of small GTPases. Thus, it is not surprising that the GEF activity is subjected to a tight and complex control (*Cherfils and Zeghouf, 2013*). Most GEFs follow the unifying mechanism of making contacts in switch I and switch II regions near the GTPase nucleotide binding pocket to facilitate nucleotide exchange. Regulation of this activity includes, but is not limited to, multiple allosteric activation sites as well as multiple domains, some of which are involved in auto-regulation. For example, one of the best studied GEFs is Sos, which activates Ras proteins. Sos is auto-regulated allosterically by the C-terminal proline-rich domain and the N-terminal Histone, Dbl-homology, and Rem domains (*Hall et al., 2002*; *Lee et al., 2017*; *Sondermann et al., 2004*; *Yadav and Bar-Sagi, 2010*).

Convergent evolution has created many structurally unrelated domains or modules capable of GEF activity (reviewed in *Cherfils and Zeghouf, 2013*). For the Rab family, which comprises the largest number of members and regulates membrane transport and organelle biogenesis, the list includes the DENN (differentially expressed in normal and neoplastic cells) domain, Vps9 (vacuolar protein sorting) domain, Sec2-domain, TRAPP (transport protein particle) complex, plus numerous heterodimeric complexes (reviewed in *Ishida et al., 2016*; *Müller and Goody, 2018*]. Out of these, GEFs containing Vps9 domains regulate diverse stages of endocytosis and early endosome transport. These GEFs have ancillary domains capable of mediating interactions with proteins and lipids, also generating layers of possible regulatory steps. Given the structural complexity of GEFs containing Vps9 domains, such as those that regulate endosomal Rabs, one would hypothesize that layers of auto-regulatory steps, such as those documented for Sos, are likely found in Vps9 domain containing GEFs.

Rabex5 is the best understood member of the Vps9 domain-containing GEFs. It has a Zn finger ubiquitin binding domain (ZnF UBD) and a motif interacting with ubiquitin (MIU), near the N-terminus (shown together in red in *Figure 1*). These domains were shown to independently bind ubiquitin molecules and binding to ubiquitinated cargo is thought to help Rabex5 localize to the plasma membrane or early endosomes (*Mattera and Bonifacino, 2008*; *Penengo et al., 2006*). A linker (teal in *Figure 1*) connects the UBDs to the rest of the protein. A 4-helix bundle (4-HB, gold in *Figure 1*) is appended to the N-terminal side of the Vps9 domain and is important for stabilizing the Vps9 domain (green in *Figure 1*) (*Delprato et al., 2004*). Together, the 4-HB and Vps9 domain make up the minimal catalytic machinery for GEF activity (*Delprato et al., 2004*). And finally, near the C-terminus is the Rabaptin5 binding site (RpBD, red in *Figure 1*) (*Delprato and Lambright, 2007*). Rabex5 exists in a tight complex with the Rab5 effector Rabaptin5, which regulates Rabex5 GEF activity (*Delprato and Lambright, 2007*; *Delprato et al., 2004*; *Horiuchi et al., 1997*; *Lippé et al., 2001a*).

Early studies of catalysis led to the proposal that Rabex5 is in an inactive state where its C-terminus hinders Rab5 binding and, thus, auto-inhibits its GEF activity (*Delprato et al., 2004*). Rabaptin5 binding to the C-terminus was suggested to cause a structural rearrangement such that the Rab5 binding to Rabex5 and subsequent catalysis can proceed (*Delprato and Lambright, 2007*; *Stenmark et al., 1995*; *Zhu et al., 2004*). However, these studies utilized truncated constructs of Rabex5 and Rabaptin5. Given its complex multi-domain organization, we reasoned that studying the full-length form of Rabex5 and its association with full-length Rabaptin5 was necessary to give a clearer picture of the auto-regulatory events. High resolution structural techniques such as x-ray crystallography, have proven difficult for full-length Rabex5 due to localized dynamics and stability problems (*Blümer et al., 2013*; *Delprato and Lambright, 2007*; *Delprato et al., 2004*; *Zhang et al., 2014*). One approach that can provide structural and mechanistic information for highly dynamic proteins not amenable to crystallography is hydrogen deuterium exchange mass spectrometry (HDX-MS).

Here, we generated full-length constructs of Rabex5 and Rabaptin5 and applied a combination of structural modeling, HDX-MS, mutagenesis and nucleotide exchange reactions to gain new insights

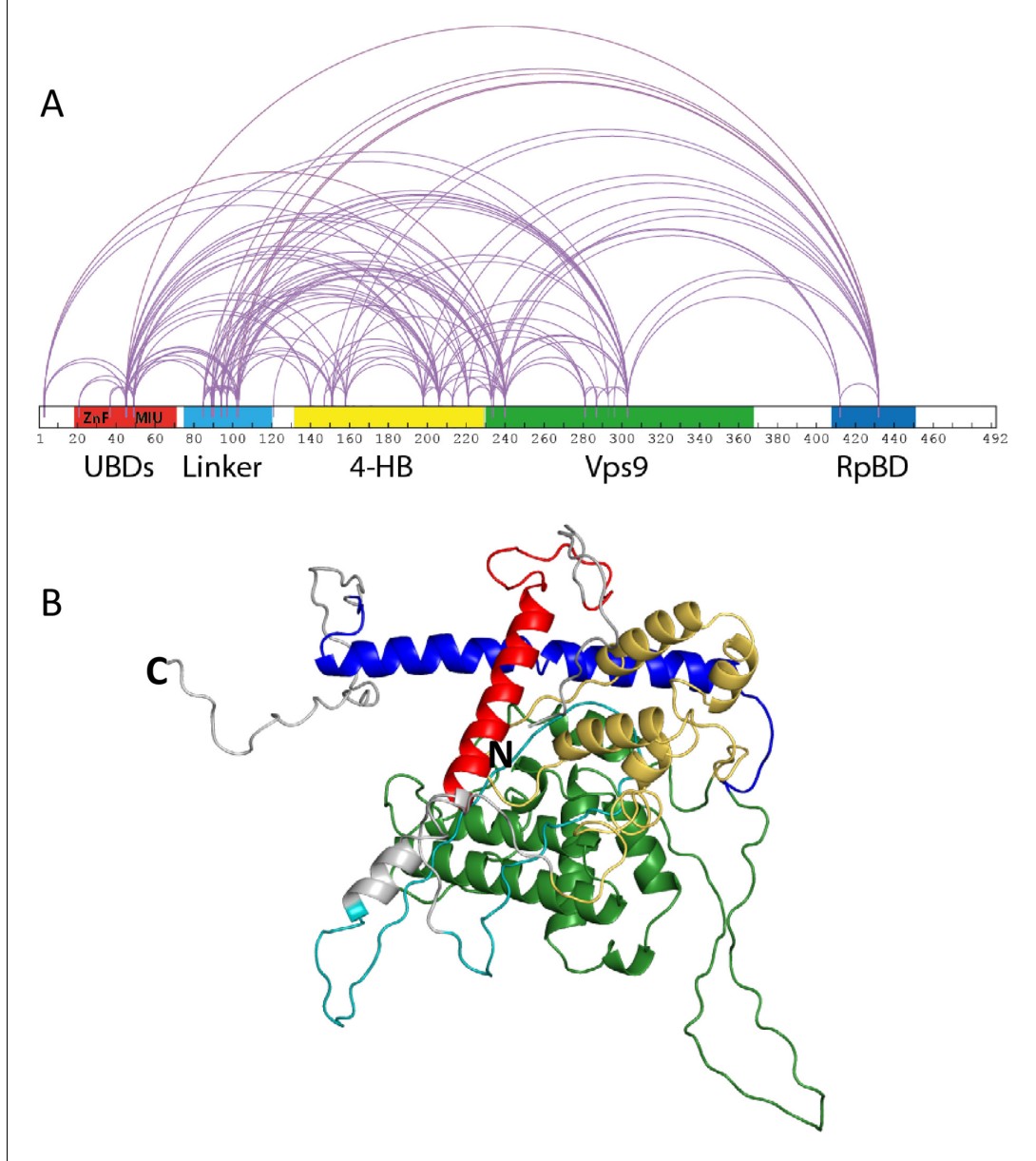

**Figure 1.** Rabex5 cross-linking-MS data and structural model. (**A**) The cross-linking-MS data are shown for apo Rabex5 and illustrated in Xwalk (*Kahraman et al., 2011*). Ubiquitin binding domains (both ZnF and MIU together in red), Linker (teal), 4-helical bundle (gold), Vps9 domain (green), and rabpatin5 binding domain (blue). (**B**) A structural model of apo Rabex5 is pseudo-colored as illustrated above. This model is one of the possible arrangements and was chosen because it was in best agreement with the cross-linking-MS and HDX-MS data.

The online version of this article includes the following figure supplement(s) for figure 1:

**Figure supplement 1.** Alignment and superimposition of our model with 1TXU.
**Figure supplement 2.** Deuterium uptake in Rabex5.

into long-range allosteric interactions. Our results revealed a previously unappreciated role for ubiquitin binding to enhance Rabex5 GEF activity and lead us to propose a new allosteric mechanism for regulating the nucleotide exchange catalytic process. They also provide evidence that dynamics within the catalytic domain are important for activity.

## Results

### A compact structure of Apo Rabex5 revealed by Cross-linking-Mass spectrometry and integrative modeling

We expressed and purified full-length Rabex5 and Rabaptin5 to study their interactions and regulation of GEF activity. We used integrative modeling to combine structural proteomics data with the available crystal structures of the 4-HB and Vps9 domain of Rabex5 to generate structural models of full-length Rabex5, one of which is shown in *Figure 1B*. As a starting point in our modeling process, we used available PDB coordinates for the UBD (blue in *Figure 1*), 4-HB (green), Vps9 domain (gold) and the C-terminal RpBD (red) as rigid body units for the modeling (see Materials and methods). Additionally, we employed beads to model flexible regions with no crystallography data. Coarse-grained docking simulations exhaustively sampled the spatial configurations of rigid domains and interconnecting flexible regions which best satisfied MS/cross-link-derived spatial restraints (see Materials and methods).

Mapping of cross-link-derived spatial restraints on available crystal structures immediately suggested that the Rabex5 structure in its Apo state better accommodates input cross-links, likely due to a different conformation of RpBD and of 4-HB domains. The observed conformational differences in relative positioning between the 4-HB and the Vps9 domain in Rabex5 structures (*Delprato and Lambright, 2007*; *Delprato et al., 2004*; *Zhang et al., 2014*), prompted us to run additional simulations allowing for rotation between the two domains forming the catalytic core. Indeed, treating the 4-HB and Vps9 domains as independent rigid bodies better satisfied cross-linking-MS data, than if kept as a single, stably folded unit (*Supplementary file 1*). This suggests that there is movement and possibly rotation of the 4-HB with respect to the Vps9 domain in solution. Next, we ran additional independent simulations additionally allowing internal flexibility of the RpBD and UBD with respect to the rest of the protein. Cluster analysis of the top scoring models identified the three best representative conformers for each simulation condition and allowed us to estimate the predicted conformations that best satisfied cross-linking-MS restraints (*Supplementary file 1*). Notably, the models in best agreement with the cross-linking-MS data (99% of cross-links satisfied) are generated by allowing flexibility between the 4-HB and Vps9 domain, as well as between the RpBD and the rest of the protein.

### The importance of the Rabaptin5 binding domain allosteric regulation of Rabex5

Given the complexity of auto-regulation for other GEFs, we sought to probe the impact of different domains of Rabex5 on the global structure of this protein, as well as its nucleotide exchange activity toward Rab5. A series of domain deletion mutants were created (*Figure 2*). The C-terminus of Rabex5 contains the Rabaptin5 binding domain (RpBD) (*Stenmark et al., 1995*; *Zhu et al., 2004*) and HDX-MS data isolated its start site to Gly407 (*Figure 1—figure supplement 2*). Deletion of that region (Gly407-Gly492) yielded the RabexΔRpBD mutant. Rabex5 contains two separate ubiquitin binding domains (UBDs) as delineated by previous crystallography experiments (*Penengo et al., 2006*). For simplicity, the Zn-finger and MIU UBDs were treated as a single UBD unit. Removal of that region created the mutant, RabexΔUBD. The Linker region connecting the UBDs to the 4-HB is dynamic and less well studied than the rest of the protein. It was unclear where to draw the boundary on the C-terminal side so two deletion mutants were created, but only one was selected for use, RabexΔ82–118 (termed RabexΔLinker). Deletion of a larger fragment (Thr82-Gln131) produced a protein which was slightly destabilized as shown by deuterium uptake and was deemed unsuitable for further use (data not shown). Finally, the RabexCAT construct was created by deleting the UBDs, the Linker, and the RpBD.

We next characterized our domain deletion mutants, by determining their nucleotide exchange activity and probing their structural alterations using HDX-MS (*Figures 3* and *4*, respectively). Consistent with precedent, the full-length Rabaptin5 protein showed a rate enhancement upon binding for WT Rabex5 and the Rabex5 mutants tested (*Delprato and Lambright, 2007*; *Delprato et al., 2004*; *Zhang et al., 2014*)(*Figure 3—figure supplement 1*). Full-length Rabex5, along with some of the mutant proteins, were somewhat unstable and difficult to express and purify in the absence of

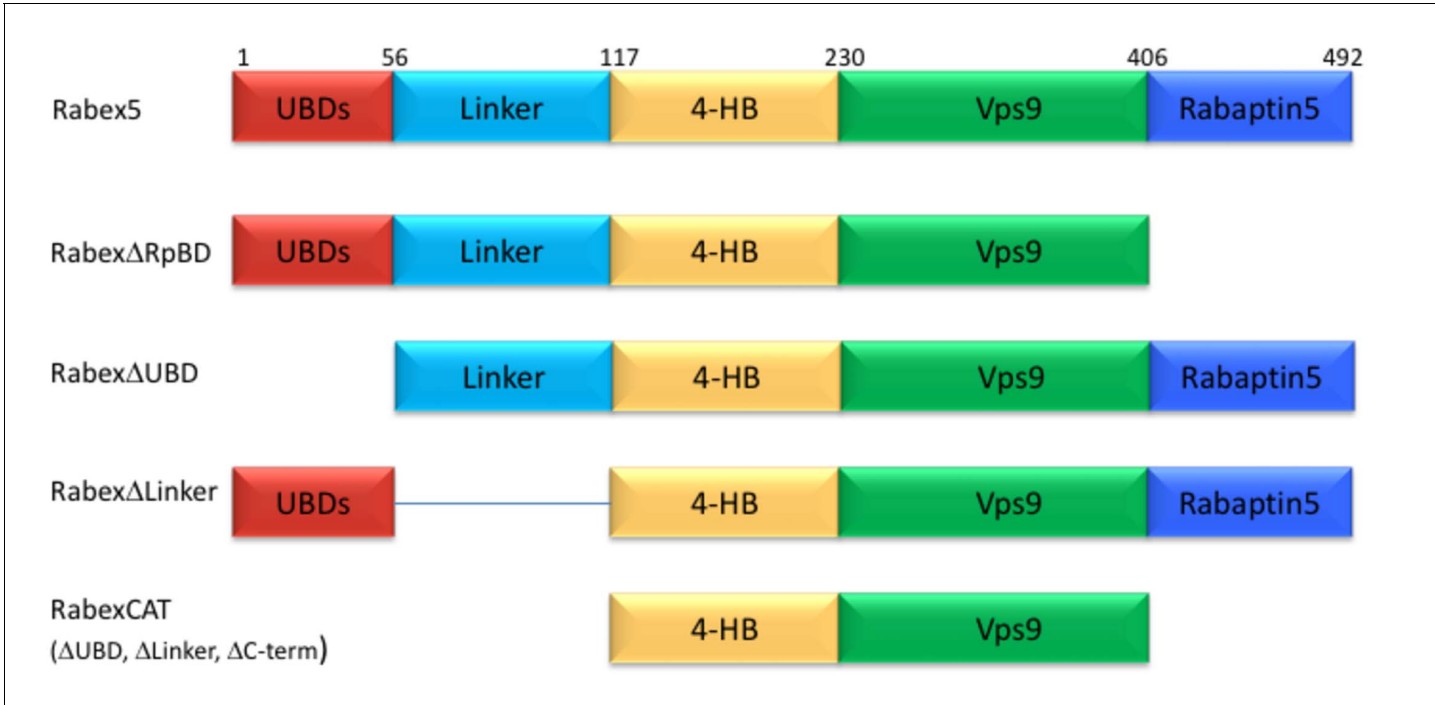

**Figure 2.** Domain deletion mutants. A series of domain deletion mutants were created as indicated. Rabex5 amino acid numbering is shown for comparison.

Rabaptin5. Thus, we compared Rabex5 domain deletion mutants only in complex with full-length Rabaptin5 (with the exception of RabexΔRpBD, which is unable to bind Rabaptin5).

It was previously suggested that the C-terminus of Rabex5 is folded over the Vps9 domain blocking the Rab5 binding site to produce auto-inhibition of nucleotide exchange activity (*Delprato and Lambright, 2007*; *Delprato et al., 2004*; *Zhang et al., 2014*). However, the cross-linking-MS data suggest otherwise. Lys433, which is located near the C-terminus within the Rabaptin5 binding domain, formed numerous cross-links with the rest of the protein. The majority of the connections were made with either the Linker or 4-HB (*Figure 1*). Cross-links were also made with Lys234, Lys241, and Lys304, which are adjacent to the Rab5 binding site within the Vps9 domain. However, these residues were also found to be cross-linked with Lys413, also near the C-terminus, suggesting substantial flexibility in the connectivity between the Vps9 domain and the C-terminus in the Rabex5 apo structure. Thus, our cross-linking-MS data suggest that in apo Rabex5, the C-terminus makes contacts mainly with amino acid residues within the Linker and 4-HB. Interestingly, the Rab5 binding site within the Vps9 domain was largely free of intra-domain cross-links, which suggests that it is mainly solvent accessible rather than occluded by any other part of the protein.

To select the best structural model, we pseudo-colored the most promising structural models using the deuterium uptake of WT-Rabex5 alone and in complex with Rabaptin5. The model in best agreement with these data was generated from 4n3z with maximum flexibility. This model will be used throughout the manuscript. Figure S4 shows the deuterium uptake after 10 s for apo Rabex5. Two important regions of the protein with helical propensity, the MIU and RpBD are shown as stable helices even though the HDX-MS data suggest they are largely flexible in the apo state, because illustrating them as helices makes them much easier to visualize. Gly407-Gly492 is highly dynamic with ~100% deuterium uptake in 10 s, (Figure S4) and thus unstructured prior to binding Rabaptin5. Upon binding Rabaptin5, one sees dramatic protection of Gly407-Glu460, as expected from the formation of a dimeric coiled-coil between the two proteins (*Figure 4A,E*). There is also mild protection extending from Asn335-Leu406, the C-terminal part of the Vps9 domain, including part of the Rab5 binding site. Unexpectedly, we found protection in the Zn-finger UBD, extending Leu18-Cys38, coupled with a very mild destabilization of the MIU UBD (Trp39-Asp54), suggesting that the binding of

Rabaptin5 serves to alter ubiquitin binding by Rabex5. Another unexpected finding is enhanced exchange by the Linker residues Phe83-Glu120 upon binding Rabaptin5 (*Figure 4A,E*).

Next, we compared the deuterium uptake of full-length Rabex5 with RabexΔRpBD, (*Figure 4B, E*). The results are virtually identical to those induced by Rabaptin5 binding to full-length Rabex5, suggesting that the allosteric structural alterations in Rabex5 are caused primarily by breaking contacts made between the RpBD and the Linker, causing the enhanced deuterium uptake in the Linker upon binding Rabaptin5 or deletion of the RpBD in Rabex5 (compare *Figure 4A,B* or compare the orange and gray traces in *Figure 4E*).

## The importance of the UBDs and Linker in allosteric regulation of Rabex5

To further dissect potential interactions between domains, we examined the RabexΔUBD mutant, which had a ~ 2 fold increase in nucleotide exchange activity (*Figure 3*), suggesting that the UBDs play a key role in the auto-regulation of Rabex5 GEF activity. Removal of the UBDs also created a destabilization of Ile339-Phe382, which encompasses part of the Rab5 binding site within the Vps9 domain (*Figure 4C,E*). This suggests that the presence and location of the UBDs stabilizes the Vps9 domain and auto-inhibits nucleotide exchange activity. Our current structural model of Rabex5 (*Figure 1*) places the UBD on the opposite side of the protein relative to the Rab5 binding site, so a *direct* interaction is not feasible. If the UBDs serve an auto-regulatory role for Rabex5, one would expect that ubiquitin binding would modulate nucleotide exchange activity in Rabex5. To test this hypothesis, we monitored the effect of ubiquitin on the nucleotide exchange activity of Rabex5. The EGF receptor is ubiquitylated via a Lys63 linkage during endocytic processing (*Haglund and Dikic, 2012*). Thus, we hypothesized that Lys63 linked tetra-ubiquitin might play a role in modulating Rabex5 localization and Rab5 activation. We found that Lys63 tetra-ubiquitin stimulated nucleotide exchange in a concentration-dependent manner (*Figure 5*). Lys48 tetra-ubiquitin (the canonical signal for proteosomal degradation; *Akutsu et al., 2016*) was also tested and produced milder GEF rate enhancement compared with Lys63 tetra-ubiquitin (data not shown). Linear di-ubiquitin did not produce rate enhancement up to 5 µM (data not shown). This suggests that ubiquitin binding could not only localize Rabex5 to an endosome containing Ubiquitylated cargo, but also enhance Rab5 GEF activity at that location, thus activating Rab5 in a cargo-specific manner.

Removal of 82–118 in RabexΔLinker caused a ~ 50% loss in nucleotide exchange activity, suggesting an unexpected role for this region in modulating catalysis (*Figure 3*). This is combined with an increase in deuterium uptake over the entire Rabaptin5 binding site (Gly407-Glu460), with the most dramatic effect localized to Met422-Glu431 (*Figure 4D,E*). Given that the structural alterations are largely limited to the RpBD, one can exclude global misfolding of this mutant leading to the decreased enzymatic activity (*Figure 4D,E*). This suggests an important and hitherto undocumented role by the Linker in modulating both nucleotide exchange and interaction with Rabaptin5. Removal of the Linker region in Rabex5 resulted in destabilization of the RpBD, but caused no detectable difference in complex formation, dimerization of the complex, or deuterium uptake in Rabaptin5 (data not shown). The cross-linking data illustrate a central location of the Linker within Rabex5. The Linker forms numerous cross-links with the MIU, 4-HB, Vps9 domain, and the C-terminal RbBD (*Figure 1*). Together these cross-links account for just over 41% of the total, while the Linker accounts for less than 5% of Rabex5, suggesting that it holds a key position in Rabex5 for mediating inter-domain communication and auto-regulation.

Much of the current understanding of the Rabex5 catalytic core structure and nucleotide exchange was derived from a construct containing Rabex132-394, because it was sufficient for catalysis while being amenable to crystallization (*Delprato and Lambright, 2007*; *Delprato et al., 2004*; *Zhang et al., 2014*). We expressed and purified this protein and characterized it by HDX-MS. The results show this construct to be substantially destabilized through the entire 4-HB and Vps9 domain compared with the full-length Rabex5 (*Figure 4—figure supplement 1*). We generated a construct similar to Rabex132-394, but with slightly different start and end points such that it aligned better with our mutants. The resulting protein, RabexCAT, was similarly destabilized compared with WT Rabex5 protein (data not shown). Its nucleotide exchange activity was roughly 2-fold higher than WT Rabex5:Rabaptin5 complex (*Figure 3*), suggesting that catalytic domain dynamics might be linked to activity.

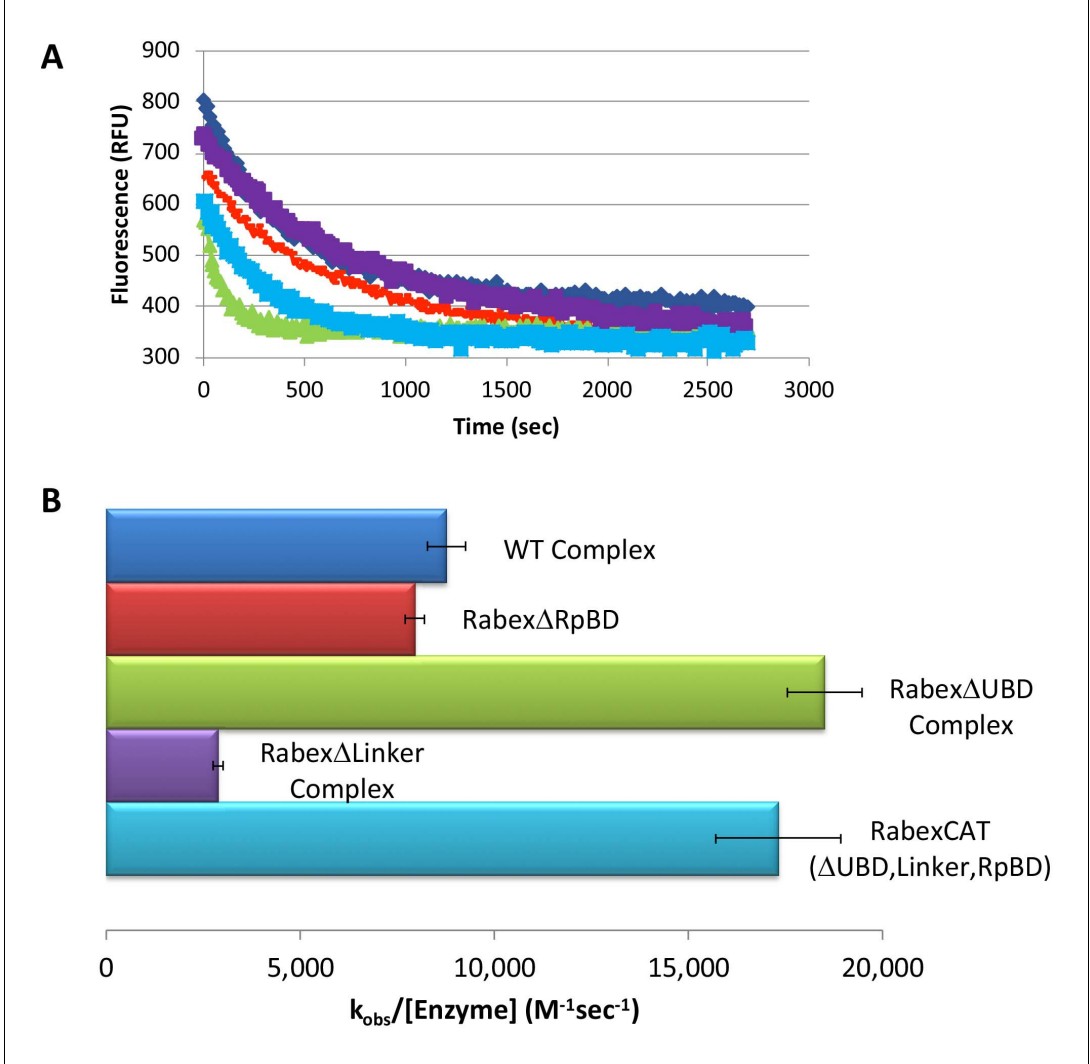

**Figure 3.** Rabex5 nucleotide exchange kinetics. Nucleotide exchange kinetics are shown for wild-type Rabex5:Rabaptin5 complex as well as the domain deletion mutants, all in complex with full-length Rabaptin5 (where applicable). (**A**) A single replicate data trace is shown. WT complex (blue diamonds), RabexΔRpBD (orange dashes), RabexΔUBD complex (green triangles), RabexΔLinker (purple squares), RabexCAT (blue squares). All enzymes were used at 0.5 µM with the exception of RabexCAT, which was used at 0.25 µM. (**B**) A compilation of nucleotide exchange kinetics data is shown. Averages were calculated from three individual experiments containing three replicates.

The online version of this article includes the following source data and figure supplement(s) for figure 3:

**Source data 1.** Source data for Rabex mutant nucleotide exchange.
**Source data 2.** Source data for Rabex mutant nucleotide exchange.
**Figure supplement 1.** Nucleotide exchange kinetics.

To delve more deeply into the process of nucleotide exchange, we monitored the deuterium uptake of Rabex5 and Rabaptin5 during the GTPγS loading process. To ensure formation of a ternary Rabex5:Rabaptin5:Rab5 complex, Rabex5:Rabaptin5 was pre-incubated with a 5-fold molar excess of GDP-bound Rab5. The mixture was subsequently incubated ±GTPγS in deuterated buffer for 60, 300 or 900 s. The Zn-finger UBD showed decreased deuterium uptake and thus was stabilized during the nucleotide exchange process (*Figure 6A,B*). Stabilization of the UBD is likely due to the release of Rab5 after nucleotide exchange, as the binding of inactive Rab5 to the Rabex5:Rabaptin5 complex causes destabilization of the Zn-finger UBD (data not shown). There is also mild destabilization of portions of the 4-HB and Vps9 domain, suggesting that nucleotide exchange is coupled with backbone motions within parts of the 4-HB and Vps9 domain. This helps explain why enhanced flexibility in parts of the catalytic core is correlated with increased nucleotide exchange, as was shown in

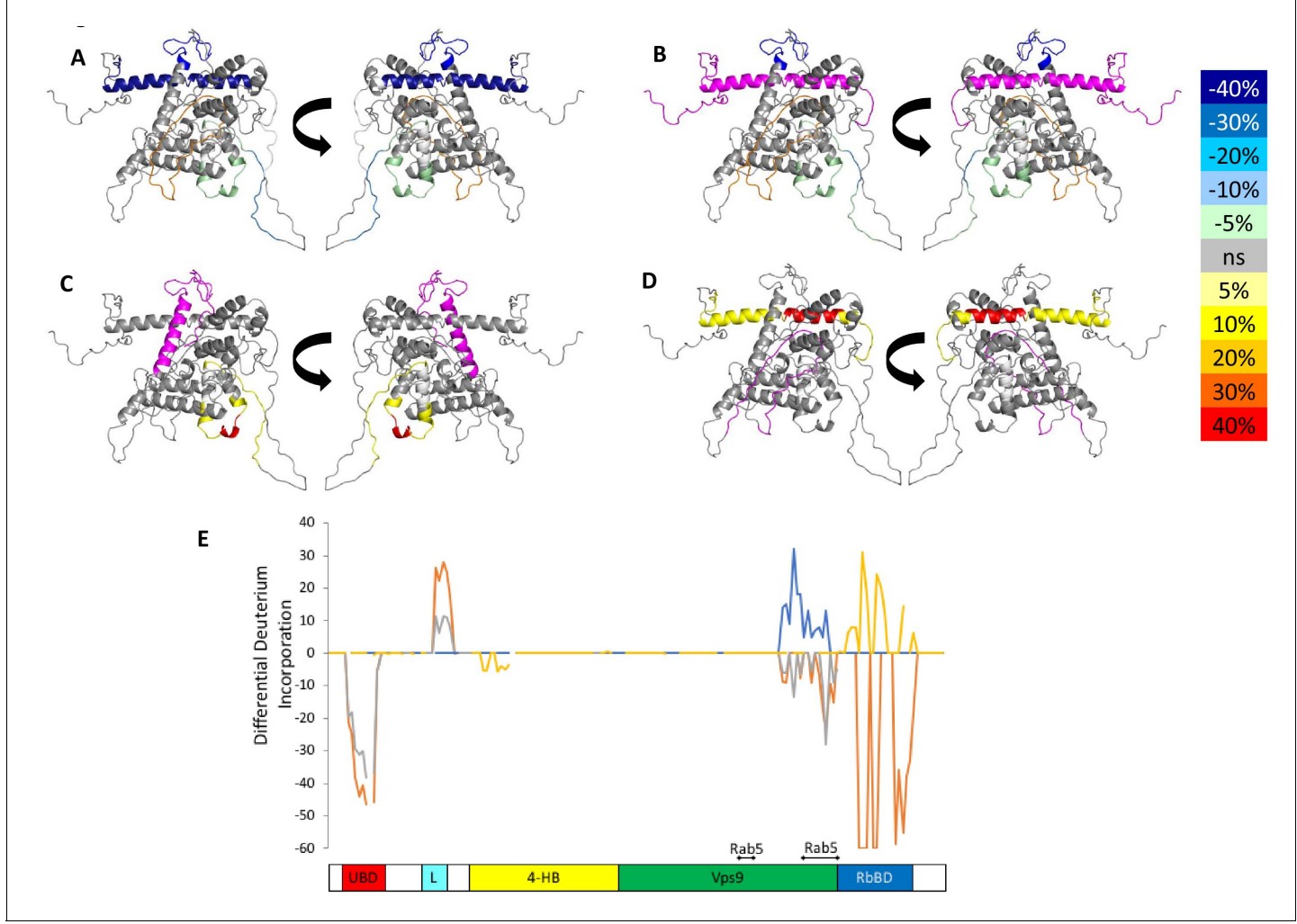

**Figure 4.** HDX-MS data. This figure shows differential uptake of deuterium. In each case, the coloring scheme is as follows: no statistically different uptake (gray), regions missing peptide coverage(white), regions deleted in mutants (magenta), regions stabilized or protected from deuterium exchange (cool colors, as shown in the figure), and regions showing enhanced exchange correlating with enhanced dynamics (warm colors, as shown in the figure). (**A**) WT Rabex5 vs Rabex5:Rabaptin5 Complex, (**B**) WT Rabex5 vs RabexΔRpBD, (**C**) WT Rabex5:Rabaptin5 Complex vs RabexΔUBD: Rabaptin5 Complex, (**D**) WT Rabex5:Rabaptin5 Complex vs RabexΔLinker:Rabaptin5 Complex. Panel **E** shows another view of the differential deuterium uptake experiments. The data from panel A (orange), panel B (gray), panel C (blue), panel D (gold). This view highlights in which domain the differences in deuterium uptake can be found as it might not be readily apparent in the model views. The Rab5 binding sites within the Vps9 domain are highlighted.

The online version of this article includes the following source data and figure supplement(s) for figure 4:

**Source data 1.** HDX-MS differential uptake values.
**Figure supplement 1.** Raw HDX-MS data.
**Figure supplement 2.** Raw HDX-MS data.
**Figure supplement 3.** Raw HDX-MS Data.
**Figure supplement 4.** Raw HDX-MS data.
**Figure supplement 5.** HDX-MS data.

the domain deletion mutants, RabexΔUBD (*Figure 4C*), RabexΔ82–132 (data not shown), Rabex132-394, (*Figure 4—figure supplement 1*), and our RabexCAT (*Figure 4—figure supplement 1*). Each of these constructs showed enhanced backbone dynamics within the catalytic core and enhanced nucleotide exchange activity compared with full-length Rabex5. Unexpectedly, during the first minute of GTPγS exchange reaction, peptides from the Linker region Glu87-Glu120 in Rabex5 disappeared followed by reappearance after 300 s (*Figure 7B and D*). The recovery of these peptides

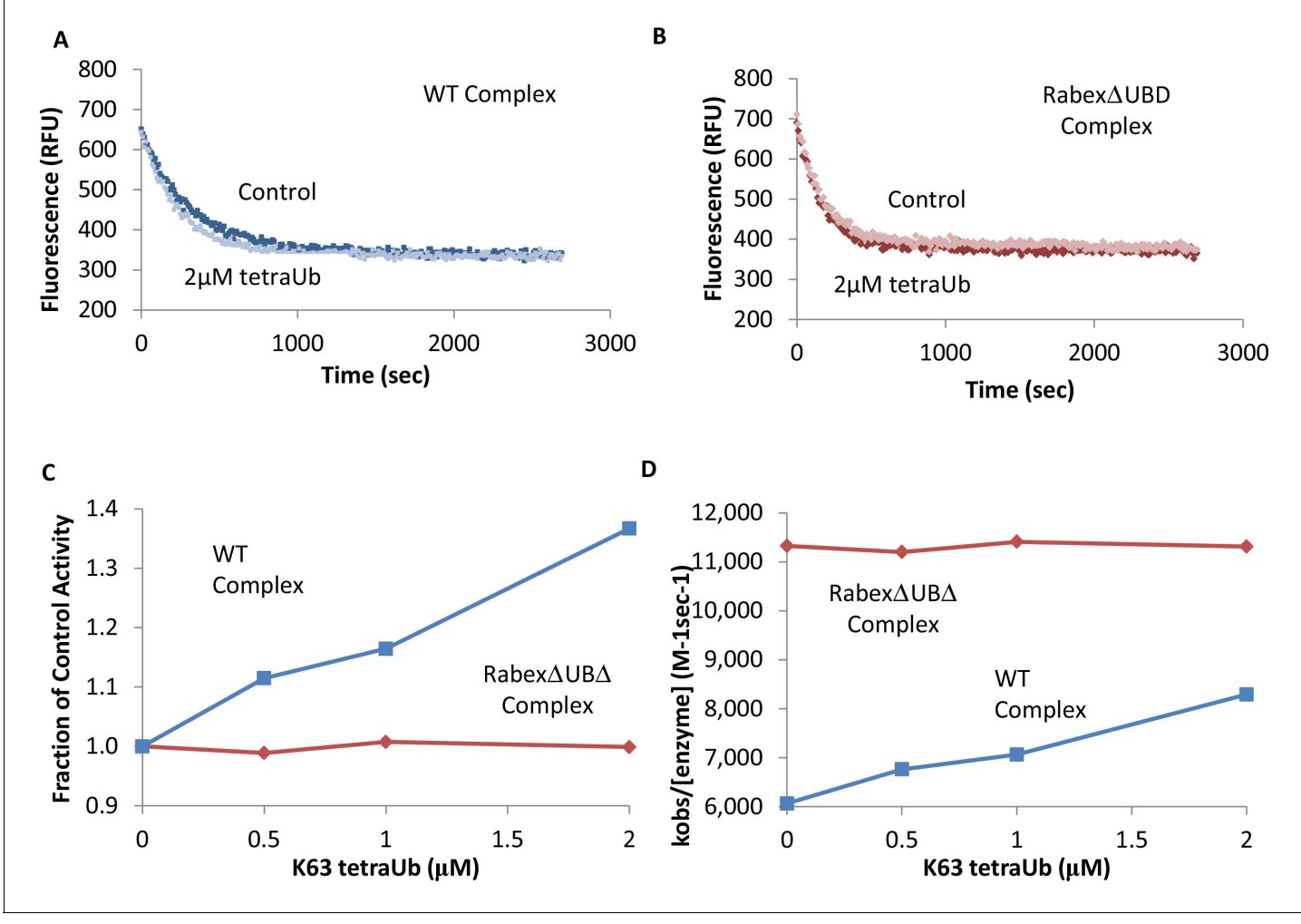

**Figure 5.** Effects of TetraUb on nucleotide exchange. Nucleotide exchange kinetics in the absence or presence of Lys63 linked tetra ubiquitin. Panel (**A**) shows an example data trace for WT Rabex5:Rabaptin5 complex alone (dark blue) and plus tetraUb (light blue). Panel (**B**) shows an example data trace for RabexΔUb:Rabaptin5 complex alone (dark rust) and plus tetraUb (light rust). Panels (**C**) and (**D**) show the average of 2 experiments each containing three replicates for WT Rabex5:Rabaptin5 complex (blue squares) and RabexΔUb:Rabaptin5 complex (rust diamonds) with varying concentrations of tetraUb.

The online version of this article includes the following source data for figure 5:

**Source data 1.** Nucleotide exchange kinetics +/- Ubiquitin.

continued in the 900 s time point (data not shown). Similar trends were seen in multiple overlapping peptides in Rabaptin5 (Leu316-Glu342) and for both proteins, only in the presence of GTPγS, suggesting this is caused by the nucleotide exchange process rather than a technical artifact. These data further support the idea that part of the Rabex5 Linker, specifically Glu87-Glu120 plays an important role in modulating Rabex5 structure as well as nucleotide exchange. Taken together, these data give us new insights and previously unrecognized roles for the UBDs as well as the Linker in the auto-regulation of Rabex5. Also highlighted is the correlation of protein dynamics within the catalytic core and catalysis.

## The Rabex5 UBD regulates the endosomal association and GEF catalytic activity in vivo

Our data suggest that ubiquitin binding enhances Rab5 GEF activity (*Figure 5*). We set to test this idea in vivo by comparing the activity of Rabex5 with that of the RabexΔUBD mutant. As a control, we expressed the RabexΔLinker which has lower catalytic activity than wt Rabex5 (*Figure 3*). For

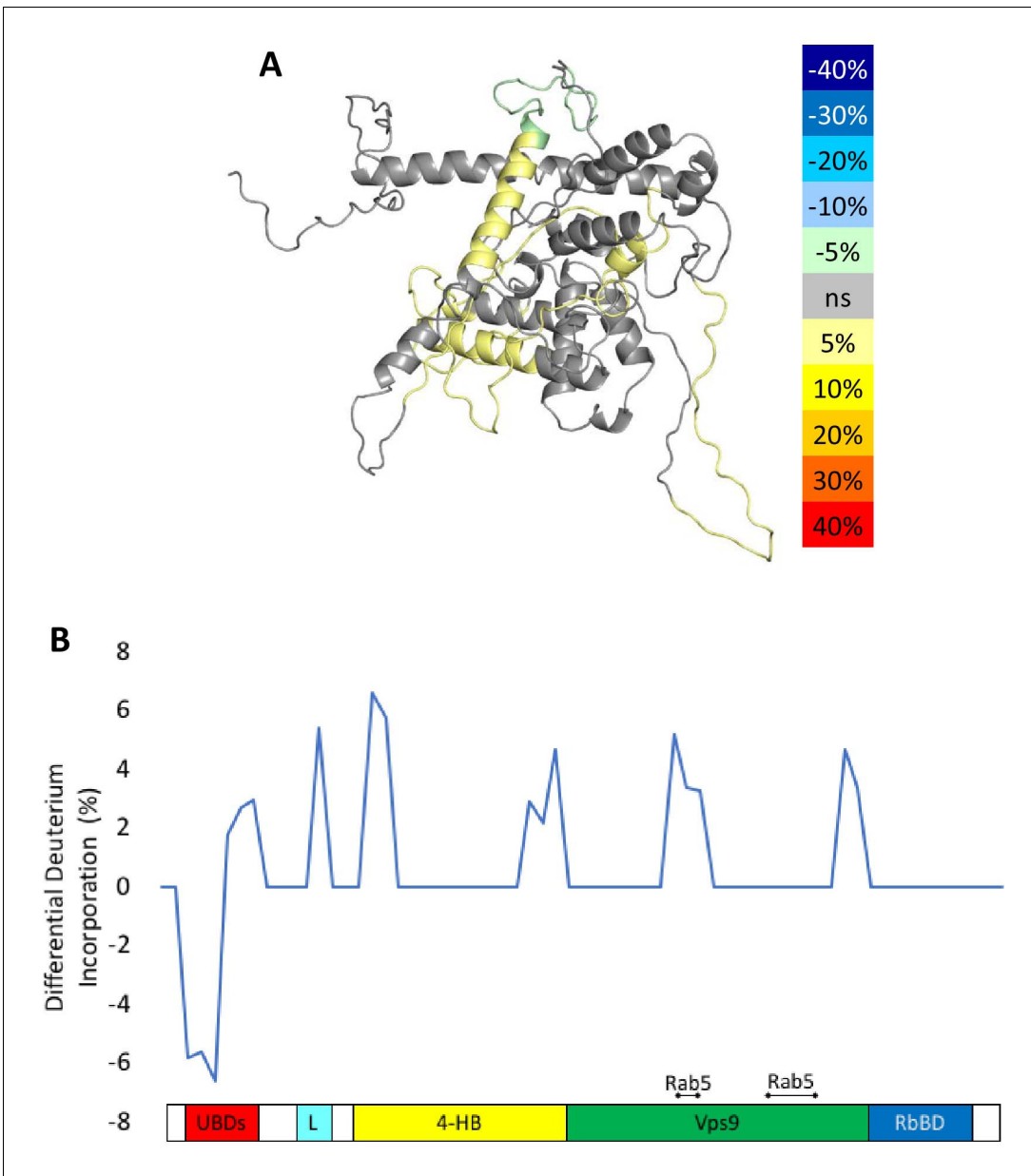

**Figure 6.** HDX-MS results during nucleotide exchange. The differential uptake of deuterium occurring during the nucleotide exchange reaction are illustrated using our structural model (**A**). In each case, the coloring scheme is as follows: no statistically different uptake (gray), regions missing peptide coverage(white), regions protected from exchange (pale green), and regions showing enhanced exchange (yellow). Panel E shows another view of the differential deuterium uptake experiments which domain the differences in deuterium uptake can be found, as it might not be readily apparent in the model view.

The online version of this article includes the following source data and figure supplement(s) for figure 6:

**Source data 1.** HDX-MS differential uptake values.
**Figure supplement 1.** Raw HDX-MS data.

this, we generated HeLa cells devoid of Rabex5 using CRISPR/Cas9 (HeLa Rabex5 KO). We expressed HA-tagged Rabex5 and mutants, and quantified the effects on the early endosomal network labelled by EEA1 and Rabaptin5 (we could not use anti-Rab5 antibodies due to incompatibilities of secondary antibodies) in relation to the levels of the expressed proteins (low, medium, high, *Figure 8*). In Rabex5 KO cells, the early endosomal network appeared altered. The most striking

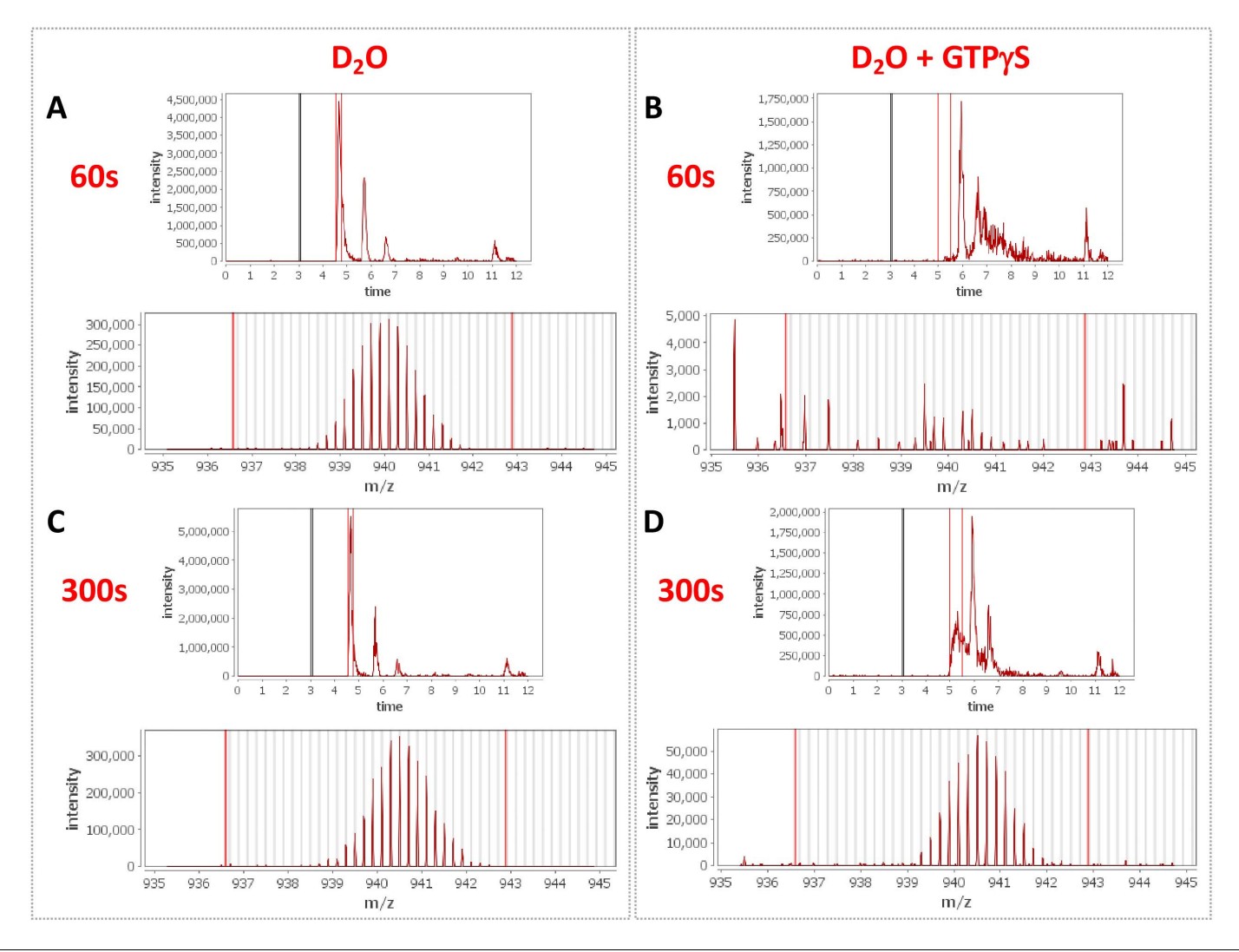

**Figure 7.** Peptide profiles during nucleotide exchange. Total ion current and centroid spectra for a peptide corresponding to Rabex5 (83-120) in the presence (**B and D**) or absence of GTPγS (**A and C**) after 60 s (**A and B**) or 300 s (**C and D**) of deuterium uptake. Note the disappearance of signal at 60 s upon addition of GTPγS and partial recovery at 300 s.

phenotype was the redistribution of early endosomes to the periphery of the cell (*Figure 8A*, compare HeLa wt with KO HeLa, see inset). The irregular shape of the EEA1-positive structures suggests that they are clusters of endosomes (*Figure 8A*, see inset), which are quantified as increase in mean endosome size (*Figure 8B*). This phenotype reflects a perturbed progression from many small endosomes at the cell periphery to fewer larger endosomes at the cell center (*Collinet et al., 2010*). Expression of Rabex5 at low levels restored the normal morphology of early endosomes, whereas higher levels of expression caused the appearance of enlarged early endosomes, typically round objects with visible lumen and located in the perinuclear region (*Figure 8A,B*, compare HeLa Rabex5 low vs high). This phenotype is consistent with a gain of function due to high Rab5 activity (*Bucci et al., 1992*). The RabexΔUBD mutant also rescued the Rabex5 KO phenotype at low expression levels and yielded a gain of function phenotype at high levels (*Figure 8A,B*, compare KO HeLa Rabex5ΔUBD low vs high), despite the fact that it was less efficiently recruited to early endosomes than Rabex5 (*Figure 8C*). This is consistent with the high constitutive GEF activity and lack of UBD contributing to the endosomal localization of this mutant. In contrast, the RabexΔLinker rescued the Rabex5 KO phenotype, but was much less potent on the stimulation of early endosome size (*Figure 8A,B*, compare KO HeLa Rabex5Δlinker low vs high), despite its localization to early

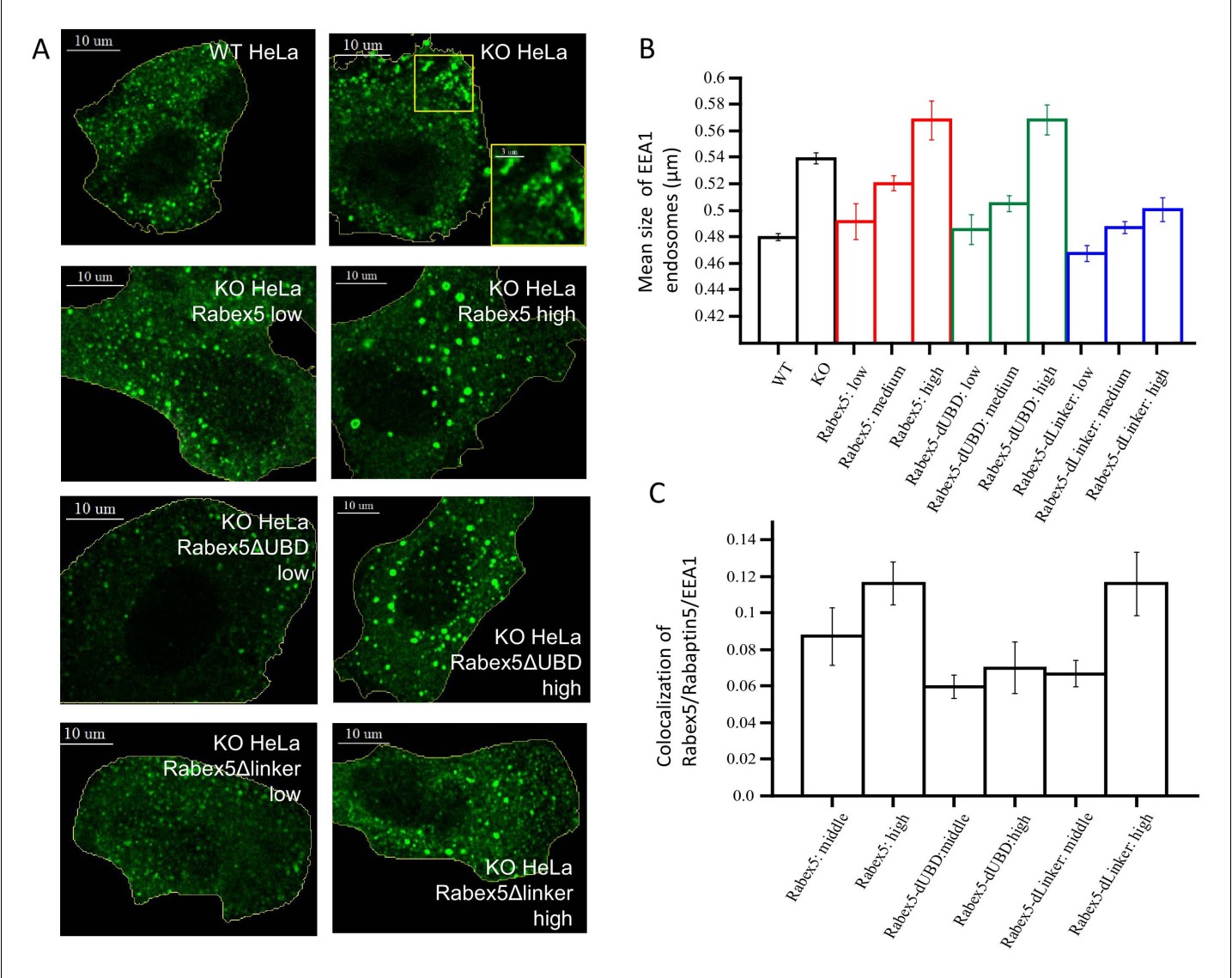

**Figure 8.** Rabex5 knock-out and rescue in HeLa cells. Confocal microscopy images of WT and Rabex5 knock-out KO HeLa cells stained with EEA1 antibodies are shown in comparison with Rabex5 KO cells expressing Rabex5, Rabex5DUBD and Rabex5DLinker (A). Images highlight cells expressing low and high levels of Rabex5 protein (not shown to better display the EEA1 endosomal pattern) and thus are masked to hide cells with other expression levels. The mean size of EEA1 positive endosomes (B) is shown for WT, Rabex5 KO cells and cells expressing the Rabex5 proteins. The colocalization of Rabex5 and EEA1 (C) is quantified to illustrate the amount of Rabex5 on the endosome and how that varies with expression level. The online version of this article includes the following figure supplement(s) for figure 8:

**Figure supplement 1.** Western blot of WT and Rabex5 knockout HeLa.

endosomes (*Figure 8C*), consistent with its lower catalytic activity. Altogether, these results indicate that the Rabex5 UBD regulates the association of Rabex5 to early endosomes and its GEF catalytic activity, as its removal leads to a high level activity despite a low association with early endosomes.

## Discussion

It has long been known that Rabex5 binds ubiqutin and its GEF activity is stimulated by Rabaptin5 (*Blümer et al., 2013*; *Delprato and Lambright, 2007*; *Horiuchi et al., 1997*; *Lippé et al., 2001b*; *Mattera and Bonifacino, 2008*). In this work, we aimed at gaining a clearer picture of how Rabex5 is regulated by studying full-length Rabex5 and Rabaptin5, retaining all relevant binding sites and

three-dimensional structure, using a structural proteomics approach including cross-linking-MS and HDX-MS. Comparisons between full-length proteins and domain deletion mutants yielded new insights that were not previously revealed with truncated proteins. We provide evidence of long-range interactions between the RpBD and the Linker, which by modulating the structure of the Vps9 domain, regulate its nucleotide exchange activity. Our results may have implications for the allosteric activation of Rab GEFs as well as GEFs for small GTPases in general. They also provide another example of how dynamics within the catalytic core of an enzyme is important for activity.

We propose a novel model to explain how allosteric modulation of Rabex5 structure regulates endosomal Rab5 activation during cargo transport. Rabaptin5 is complexed to Rabex5 and this inter-action is required to confer stability and allosteric regulation on Rabex5 catalytic GEF activity. Unexpectedly, from our analyses emerged a more important role of the UBD in modulating such activity. It was previously proposed that ubiquitin binding serves to localize Rabex5 on the endosomal membrane (*Blümer et al., 2013*; *Mattera and Bonifacino, 2008*; *Penengo et al., 2006*). Our results indicate that binding of ubiquitin to the UBD of Rabex5 goes beyond mere recruitment as it also enhances nucleotide exchange activity. Deletion of the UBDs causes at least a 2-fold increase in nucleotide exchange as well as enhanced deuterium uptake in a portion of the Rab5 binding site. The HDX-MS results for the C-terminal deletion mutant of Rabex5, RabexΔRpBD or the binding of full-length Rabex5 to Rabaptin5 via the C-terminus both alter the structure of the UBDs. This long-range allosteric regulation as well as enhancement of Rabex5 GEF activity by ubiquitin binding has never before been revealed and suggests a major role for ubiquitin in regulating Rabex5. This implies that endocytosis of ubiquitinated cargo, would supply binding sites for Rabex5 or the Rabex5:Rabaptin5 complex, localizing as well as enhancing Rab5 activation on early endosomes. Studies in vivo showed that expression of Rabex5ΔUBD efficiently rescued loss of Rabex5 despite a reduced association with early endosomes. These results are consistent with the higher GEF catalytic activity of the mutant that compensates for the lower endosomal recruitment. Rabaptin5 also contributes to the endosomal recruitment of Rab5. Being a Rab5 effector, Rabaptin5 can bind active Rab5, stabilizing it on the endosomal membrane (*Horiuchi et al., 1997*; *Stenmark et al., 1995*). This serves to create a positive feedback loop (*Horiuchi et al., 1997*), whereby stimulation of Rab5 activity would enhance early endosome fusion, regulating the life-time of endocytosed receptors in early endosomes (*Villaseñor et al., 2015*). This is consistent with the long observed increase in early endosome fusion following internalization of EGFR (*Benveniste et al., 1989*; *Roberts et al., 2000*; *Sorkin et al., 2000*) in addition to micropinocytosis (*Argenzio et al., 2011*; *Balaji et al., 2012*; *Horiuchi et al., 1997*; *Penengo et al., 2006*; *Sönnichsen et al., 2000*). A study of the evolution of the Vps9 domain showed three independent instances of acquisition of structurally diverse UBDs: the ZnF in mammalian Rabex5, the CUE domain in yeast, and a UBD in the amebozoan *Sexangularia sp.* (*Herman et al., 2018*). The fact that Vps9 domain containing proteins had such a strong tendency to acquire UBD through independent means suggests a critical role for UBDs in regulating Vps9 activity.

The availability of full-length proteins was of fundamental importance to obtain insights into Rabex5 structure and regulation. For example, the interaction site between Rabex5 and Rabaptin5 was localized to Gly407-Glu460 in Rabex5 and Met563-Leu658 in Rabaptin5, with the core and most dramatic protection localized to Val600-Leu633. This differs slightly from the structure portrayed by PDB:4Q9U, which shows interaction between Pro392-Ile453 in Rabex5 and Phe584-Arg635 in Rabaptin5 (*Zhang et al., 2014*). The constructs used to generate PDB:4Q9U were truncated to enhance crystallization. While the construct Rabaptin5 (552-642) binds Rabex5, the exact region making contacts differs from that observed here using the full-length protein. The most dramatic difference between our results and those of PDB:4Q9U, 4N3Y, and 4N3Z are that they display an arrangement of Rabaptin5:Rabex5 (2:1) (*Zhang et al., 2014*). Size exclusion coupled with static light scattering results show a 1:1 arrangement (data not shown). Since the structures reported in *Zhang et al. (2014)* were generated with a Rabaptin5 peptide covering 552–642, one can hypothesize that the differences are caused by a non-physiological pairing of the coiled-coil with such a limited portion of Rabaptin5.

One other potentially important difference between our results and those derived from crystallography is the relative positioning of the 4-HB with respect to the Vps9 domain. The relative positions of these domains in 4Q9U differ slightly from that of 4N3Z. Our modeling simulations generated structures in better agreement with cross-linking-MS data when flexibility was allowed

between the 4-HB and Vps9 domain (see *Figure 1—figure supplement 1* for a comparison of our model (dark green and yellow) with that of 1TXU (light green and yellow) illustrating relative positions of the 4-HB and Vps9 domains). This suggests that flexibility or possibly rotation between the domains occurs in solution. Is this rotation merely the normal 'breathing' which occurs in proteins, or is it a relevant part of the nucleotide exchange mechanism? The HDX-MS results obtained during catalysis by the Rabex5:Rabaptin5 complex showed enhanced deuterium exchange occurring in the catalytic core of Rabex5 (*Figure 6A and B*). Specifically, parts of 4-HB and Vps9 were affected, leaving us to postulate that if increased backbone dynamics within the catalytic core occur during nucleotide exchange, anything restricting mobility in these regions could decrease GEF activity and, conversely, anything enhancing mobility could increase it. The domain deletion mutants illustrated that deletions which caused enhanced deuterium uptake (enhanced mobility or flexibility) within the 4-HB and/or Vps9 domain correlated with increased nucleotide exchange activity for Rab5. This is an entirely new way of thinking about Rabex5 auto-regulation. One should note that the Rab5 binding site is located on the opposite side of the Vps9 domain compared with the 4-HB (approximate Rab5 binding site is near the bottom of *Figure 1B*). Thus, rotation or repositioning of the 4-HB and/or Vps9 domain is unlikely to alter GEF activity by altering Rab5 binding directly. A number of enzymes have shown that motions detected during catalysis are also present in the apo enzyme (*Eisenmesser et al., 2005*; *Henzler-Wildman and Kern, 2007*), leading to the hypothesis that sampling of conformational sub-states observed during catalysis in the apo enzyme are a general paradigm for enzymes (*Henzler-Wildman and Kern, 2007*). Herein, we provide another example of such activity.

Another previously unknown phenomenon in Rabex5 is the communication between the C-terminus and UBDs, which seems to be mediated by the Linker. This is not the first example of a Linker being more than just a flexible tether between domains within a GEF (*Cherfils and Zeghouf, 2013*). The release of the Linker upon binding Rabaptin5 (*Figure 4A*, or 4E orange trace) suggested a critical connection between the Linker and the RpBD. Our results show an unequivocal connection between these regions since the RabexΔLinker mutant displayed enhanced deuterium uptake in the RpBD and the RabexΔRpBD mutant displayed enhanced deuterium uptake in the Linker (*Figure 4C, D* or *Figure 4E* blue and gold traces). Careful inspection of our molecular model of Rabex5 shows that the Linker is located in a critical position between the RpBD, 4-HB, Vps9 domain, and UBDs. Thus, one can see how moving the Linker can alter the entire protein and regulate GEF activity. Sequence alignment of Rabex5 with its two most similar Vps9 domain containing proteins, Gapex5 and Varp5, show that Ala123 and Pro124 within the Linker region are strictly conserved. Deletion of Thr82-Gln131 caused unexpected destabilization of the 4-HB, which was not present when only Thr82-Ile118 was deleted. Also, Rabex132-394 and RabexCAT were found to be substantially destabilized compared with WT Rabex5 (*Figure 4—figure supplement 1*). The HDX-MS results suggest that a portion of the Linker, specifically Gln119-Gln131 stabilizes and regulates the catalytic core. Sequence conservation suggests Ala123 and Pro124 could be mediating this activity.

In summary, our results show that auto-regulation of Rabex5 GEF activity is more sophisticated than previously appreciated. Using the full-length protein, as well as domain deletion mutants, helped us reveal some of the depth of auto-regulation of Rabex5. Specifically, the Linker and RpBD show correlated behavior suggesting direct contact. If this interaction is modulated, allosteric structural alterations in the UBD and Vps9 domain as well as enhanced nucleotide exchange follow. Also, the binding of ubiquitin enhanced catalysis, suggesting that ubiquitinated proteins play an important role in regulating Rabex5 and controlling endosomal trafficking. Our results correlating enhanced flexibility within the catalytic core with increased nucleotide exchange activity suggest an additional means of regulating nucleotide exchange. These results reveal layers of auto-regulation not previously suggested for Rabex5 and suggests that other GEFs may also contain as yet unappreciated regulatory mechanisms.

## Materials and methods

### Expression and purification of recombinant proteins

The gene sequences corresponding to full-length bovine Rabex5 (1-492) and full-length human Rabaptin5 (1-862) were subcloned into 6x-His or GST-containing pOEM derived vectors for

baculoviral expression. Each vector contains an HRV 3C-cleavage site to remove purification tags from the desired protein. SF+ cells growing in ESF921 media (Expression Systems) are transfected with plasmid and in-house prepared bacmid DNA. Conditioned media containing virus is harvested and used to infect SF+ cells at 1% vol/vol. Cells are harvested after 40–48 hr and frozen. Human Rab5a in a pGEX-5x vector is transformed in BL21(DE3) cells. Protein expression is induced by 0.1 mM isopropyl-1-thio-b-D-galactopyranoside to the culture. Cells are harvested after 18 hr at 16 °C and frozen. All cell pellets are resuspended in 40 mL of Rab5 buffer (20 mM tris pH 7.4, 150 mM NaCl, 5 mM $MgCl_2$, 0.5 mM TCEP) plus protease inhibitor cocktail (chymostatin 6 µg/mL, leupeptin 0.5 µg/mL, antipain-HCl 10 µg/mL, aprotinin 2 µg/mL, pepstatin 0.7 µg/mL APMSF 10 µg/mL) and lysed by sonication. Lysates were clarified by centrifugation and applied to Ni-NTA resin in the presence of 20 mM imidazole or GS-4B resin for 2 hr at 4 degC, followed by stringent washing. While on GS-resin, Rab5 is washed with Rab buffer containing 10 mM EDTA to remove endogenous nucleotide and returned to Rab buffer for cleavage from the resin. To generate GTPγS-Rab5, GS-bound protein is washed with Rab buffer containing 10 mM EDTA and 1 mM GTPγS and subsequently returned to Rab5 buffer for cleavage from the resin. All proteins are incubated with HRV 3C protease (Rabex5 and Rabaptin5) or Factor $X_a$ (Rab5) overnight at 4 °C with mixing and further purified by size exclusion chromatography using S200 or Superose six columns. Concentrations are determined using a BCA assay (Thermo Scientific) and proteins are stored at −80 degC.

## Chemical cross-linking combined with mass spectrometry

To provide distance constraints for modeling of Rabex5, cross-linking-MS has been applied according to protocols recently published in *Guaitoli et al. (2016)*. Briefly, in order to obtain a cross linker to protein ratio of 12.5:1 or 25:1, NHS-ester–based chemical cross-linking was performed to cross-link Lys residues by adding DSS H12/D12 or DSG-H6/D6 solution (both at a 12.5 mM stock solution in DMSO) to 80 µg of purified protein in HEPES-based buffer. The reactions were carried out under constant rotation at RT for 30 min. Reaction was then quenched adding Tris·HCl (pH 7.5) solution to a final concentration of 10 mM and incubated again for another 15 min at RT under constant shaking. After quenching, protein samples were precipitated using chloroform and methanol. Proteolysis was performed, and the resulting peptide mixtures were separated by size exclusion chromatography to enrich cross-linked peptides. For mass spectrometric analysis, SpeedVac-dried SEC-Fractions were re-dissolved in 0.5% TFA and analyzed by LC-MSMS using a nano-flow HPLC system (Ultimate 3000 RSLC, Thermo-Fisher) coupled to an Orbitrap Fusion (Thermo-Fisher) tandem mass spectrometer. For identification, cross-linked peptides were separated and analyzed by a data-dependent approach acquiring CID MSMS spectra of the 10 most intense peaks (TOP 10), excluding single and double charged ions. Prior to analysis via xQuest/xProphet (V2.1.1), MSMS spectra were extracted from the RAW files using ProteoWizard/msconvert (3.0.6002). The identification of monolinks, loop-links and cross-links was done based on the identification of DSS H12/D12 or DSG H6/D6 pairs. For xQuest/xProphet, standard parameters were used with methionine oxidation as variable modification. For DSS-D12 and DSG-D6, isotope differences of 12.075321 Da and 6.03705 Da were used, respectively. The parent ion tolerance was set to 10 ppm (MS) and the fragment ion tolerance to 0.3 Da (MSMS). Only those cross-linked peptides were considered for modeling which fulfilled the following minimal criteria: ID-score >28, deltaS <0.95, FDR < 0.05. Additionally, the MSMS spectra were evaluated by manual inspection to ensure a good representation of the fragment series of both cross-linked peptides.

## Structural modeling

We generated a structural input model of full length Rabex5 in its apo state using available x-ray structures of Rabex5 (PDB ID: 4N3Z and 2C7N). In order to achieve a full-length model, we created linkers joining the N-terminal helix bound to ubiquitin (PDB ID: 2C7N) to the 4-HB, Vps9 and C-terminal domains (PDB ID: 4N3Z) through Modeler (*Sali and Blundell, 1993*). A total of 100 models were generated by randomizing the amino acid Cartesian coordinates in the initial model. The twenty models with the lowest number of stereochemical constraints violations were selected and ranked according to the DOPE score (*Shen and Sali, 2006*) The best ten models were validated for their stereochemical quality through the Molprobity tool (*Chen et al., 2010*) available in the Phenix software package (*Adams et al., 2011*) and the best scoring model was retained for further

modeling steps. We also derived through the same strategy an alternative starting conformation by considering Rabaptin-bound Rabex structure (PDB ID: 4Q9U).

## Determining the Rabex in the APO state with integrative modeling platform (IMP)

We predicted the 3D structure of full length Rabex in its apo state by using cross-linking/MS-derived distance restrained docking calculations through the Integrative Modeling Platform (IMP) (*Russel et al., 2012*) package, release 2.9.0. We employed the Python Modeling Interface (PMI), adapting the scripted pipeline of a previously described procedure (*Chen et al., 2016*) and consisting in the following key steps: (1) gathering of data, (2) representation of domains and/or subunits and translation of the data into spatial restraints, (3) configurational sampling to produce an ensemble of models that optimally satisfies the restraints, and (4) analysis and assessment of the ensemble.

### System representation

The full-length Rabex initial models were used as input structures for IMP calculations. Domains of Rabex structures were represented by beads arranged into either a rigid body or a flexible string on the basis of the available crystallographic structure. We probed as input for simulation both the apo state (PDB ID: 4N3Z), as well as the Rabaptin/Rab5 bound (PDB ID: 4Q9U) Rabex x-ray structures. We also included the structure of the ubiquitin bound N-terminal helix (PDB ID: 2C7N). The beads representing a structured region were kept rigid with respect to one another during configurational sampling (i.e. rigid bodies). The following regions have been defined as rigid body entities: N-terminal $\alpha$-helix (NtH):18–71; 4 Helical Bundle (4-HB): 133–228; Vps9: 231–368; C-terminal $\alpha$-helix (CtH): 408–452. The ubiquitin structure (residues: 1–73) was considered as a single rigid body unit together with the NtH. Protein segments without a crystallographic structure or the linkers between the rigid domains were represented by a flexible string of beads, where each bead corresponded to a single residue. For both starting configurations, we performed two simulation sets: one where the 4-HB and Vps9 domains were kept as a single rigid body and the other where they were kept as independent rigid body (i.e. introducing flexible beads at residues 229–230).

### Bayesian scoring function

The cross-linking data were encoded into a Bayesian scoring function that restrained the distances spanned by the cross-linked residues (*Erzberger et al., 2014*). The Bayesian approach estimates the probability of a model, given information available about the system, including both prior knowledge and newly acquired experimental data (*Erzberger et al., 2014*; *Rieping et al., 2005*). Briefly, using Bayes' theorem, we estimate the posterior probability p(M D,I), given data D and prior knowledge I, as p(M D, I) $\propto$ p(D M, I)p(M, I), where the likelihood function p(D M,I) is the probability of observing data D, given I and M, and the prior is the probability of model M, given I. To define the likelihood function, one needs a forward model that predicts the data point (i.e. the presence of a cross-link between two given residues) given any model M and a noise model that specifies the distribution of the deviation between the observed and predicted data points. To account for the presence of noisy cross-links, we parameterized the likelihood with a set of variables $\{\psi\}$ defined as the uncertainties of observing the cross-links in a given model (*Erzberger et al., 2014*; *Robinson et al., 2015*). A distance threshold of 20 Å was employed to model DSSO cross-linkers.

### Sampling model configurations

Structural models were obtained by Replica Exchange Gibbs sampling, based on Metropolis Monte Carlo sampling (*Rieping et al., 2005*). This sampling was used to generate configurations of the system as well as values for the uncertainty parameters. The Monte Carlo moves included random translation and rotation of rigid bodies (4 Å and 0.03 rad, maximum, respectively), random translation of individual beads in the flexible segments (5 Å maximum), and a Gaussian perturbation of the uncertainty parameters. A total of 500,000 models per system were generated, starting from 100 random initial configurations and sampling with temperatures ranging between 1.0 and 2.5. We divided this set of models into two ensembles of the same size to confirm sampling convergence (data not shown).

## Analysis of the model ensemble

The 200 best scoring models (i.e. solutions) for each docking run were clustered to yield the most representative conformations. For each ensemble, the solutions were grouped by k-means clustering on the basis of the r.m.s. deviation of the domains after the superposition of the Vps9 domain (*Figure 1*). The precision of a cluster was calculated as the average r.m.s. deviation with respect to the cluster center (i.e. the solution with the lowest r.m.s. deviation with respect to the others).

For each cluster, we calculated the number of satisfied MS/cross-links by measuring the number of C$\alpha$ pairs, corresponding to cross-linked Lys, whose distance was shorter than 34 Å. We recorded the fraction of satisfied cross-links, given by the number of satisfied cross-links over the total, for all the cluster members. We finally reported the fraction of satisfied cross-links for the best scoring solution, as well as the maximum fraction obtained for a single cluster conformer and the aggregate fraction obtained by considering all the cluster members together.

For visualization purposes, we generated all atoms models by starting from the C$\alpha$ traces generated from IMP and using the *automodel()* function from Modeler (*Sali and Blundell, 1993*).

## Hydrogen deuterium exchange mass spectrometry

HDX-MS was performed essentially as previously described (*He et al., 2015*; *Mayne et al., 2011*; *Walters et al., 2012*). Proteins (1 µM) are diluted 6:4 with 8M urea, 1% trifluoroacetic acid, passed over an immobilized pepsin column (2.1 mm x 30 mm, ThermoFisher Scientific) in 0.1% trifluoroacetic acid at 15 degC. Peptides are captured on a reversed-phase C8 cartridge, desalted and separated by a Zorbax 300 SB-C18 column (Agilent) at 1 °C using a 5–40% acetonitrile gradient containing 0.1% formic acid over 10 min and electrosprayed directly into an Orbitrap mass spectrometer (LTQ-Orbitrap XL, ThermoFisher Scientific) with a T-piece split flow setup (1:400). Data were collected in profile mode with source parameters: spray voltage 3.4kV, capillary voltage 40V, tube lens 170V, capillary temperature 170degC. MS/MS CID fragment ions were detected in centroid mode with an AGC target value of $10^4$. CID fragmentation was 35% normalized collision energy (NCE) for 30 ms at Q of 0.25. HCD fragmentation NCE was 35 eV. Peptides were identified using Mascot (Matrix Science) and manually verified to remove ambiguous peptides. For measurement of deuterium uptake, 10 µM protein is diluted 1:9 in Rab5 buffer prepared with deuterated solvent. Samples were incubated for varying times at 22 °C followed by the aforementioned digestion, desalting, separation and mass spectrometry steps. The intensity weighted average m/z value of a peptide's isotopic envelope is compared plus and minus deuteration using the HDX workbench software platform. Individual peptides are verified by manual inspection. Data are visualized using Pymol. Deuterium uptake is normalized for back-exchange when necessary by comparing deuterium uptake to a sample incubated in 6M urea in deuterated buffer for 12–18 hr at room temperature and processed as indicated above.

## Nucleotide exchange kinetics

Nucleotide exchange kinetics were measured by monitoring the release of the mant-GDP nucleotide analog, 2′-(3′)-bis-O-(N-methylanthraniloyl)-GDP (Jena Biosciences). Rab5 was loaded with mant-GDP in Rab5 buffer + 10 mM EDTA and 5-fold excess mant-GDP. Rab5(mant-GDP) was separated from unbound nucleotide by size exclusion chromatography using Rab5 buffer. Samples were excited at 355 nm and the emission monitored at 448 nm. Reactants were mixed in 20 mM tris (pH 8), 150 mM NaCl, 0.5 mM MgCl$_2$. Exchange reactions were initiated by addition of 200 µM GTP. Data were collected using a Spark microplate spectrophotometer (Tecan). For the enzyme concentrations used herein, some of the reactions were past the initial linear phase of product release. To minimize experimental error, observed pseudo-first order rate constants ($k_{obs}$) were calculated in Prism using the one phase decay non-linear fit. The catalytic efficiency was obtained by subtracting the intrinsic rate ($k_{intr}$) of GDP release and dividing by the concentration of Rabex5.

## Rabex5 knockout and rescue

The knockout was generated as described (*Spiegel et al., 2019*) by deleting the first common exon to all annotated isoforms. Two pairs of CRISPR guide RNA (cr1/cr3, cr2/cr4) flanking the Rabex exon were selected based on low off-target and best on-target activity using http://crispor.tefor.net. The guideRNAs were ordered as crRNA from Integrated DNA Technologies (IDT). The result was a knock

out of the rabex5 gene on two different alleles, that resulted from a deletion between the CRISPR guides of 640 bp and a bigger deletion of 930 bp; both deleting the targeted exon. Homozygous knockout required two rounds as the first resulted in heterozygous knockout. The Rabex knock-out was validated by sequencing and Western blot (*Figure 8—figure supplement 1*).

The HeLa Kyoto and derived cell lines were grown in DMEM supplemented with Penicillin (100 units/mL), Streptomycin (100 µg/mL) and Fetal Bovine Serum (10%). Cell lines were authenticated using STR profiling and tested negative for mycoplasma. The codon-optimized human Rabex sequence was obtained from GenScript Biotech. The different Rabex variants were C-terminally tagged with GGGGSGGGGS-HA (hRabex5-GS-HA, hRabex5 delta 82–117-GS-HA, hRabex5 56–492-GS-HA) through amplification by PCR and subsequently cloned into the pCMV5 Flag-Smad1 by replacing the Smad gene between BglII and SalI. Rescue experiments were performed by transfecting 1 µg of plasmid construct expressing Rabex variants using the Lipofectamine 3000 reagent following the manufacturers protocol into either HeLa Kyoto WT or the Rabex KO grown on glass coverslips. 48 hr post-transfection, cells were fixed with 4% (w/v) paraformaldehyde in PBS for 15 min at 37°C and subsequently permeabilized with 0.1% (w/v) Triton X-100 in PBS for 5 min. Samples were blocked with 3% (w/v) BSA in PBS for 45 mins at RT, followed by incubation with primary antibodies diluted in 3% (w/v) BSA in PBS for 1 hr at room temperature (in-house-made ABs: rabbit-anti-EEA1 antibodies, 1:1000; anti-Rabaptin5 antibodies, 3 µg/mL; anti-HA-tag antibodies 1:200). After washing, cells were incubated for 30 mins with an appropriate secondary antibody conjugated with an Alexa-dye (Invitrogen), 1:500. Coverslips were mounted in ProLong Diamond (ThermoFisher) cured for at least 24 hr and subsequently sealed with nail polish. Cells were imaged using a Zeiss LSM 880 Airy upright (Zeiss) microscope equipped with a 63x/1.4 Plan-Apochromat, Oil, DIC (Zeiss) objective. All images were acquired under equal conditions. Images were analyzed using the lab custom built software Motion Tracking, refer to the website for downloading (http://motiontracking.mpi-cbg.de/get/).

## Acknowledgements

We warmly thank Anglika Giner, Ramona Schäfer and the Core Facility for Bioanalytics at the University of Tübingen for technical assistance as well as the Mass Spectrometry, Light Microscopy, Antibody, and Genome Editing Facilities at Max Planck Institute of Molecular Cell Biology and Genetics for access to their facilities and technical support.

## Additional information

### Funding

| Funder | Grant reference number | Author |
|---|---|---|
| Max-Planck-Gesellschaft | Open-access funding | Marino Zerial |
| Deutsche Forschungsgemeinschaft | Project Number 112927078 - TRR 83 | Marino Zerial |

The funders had no role in study design, data collection and interpretation, or the decision to submit the work for publication.

### Author contributions

Janelle Lauer, Conceptualization, Data curation, Formal analysis, Investigation, Methodology, Writing—original draft, Project administration, Writing—review and editing; Sandra Segeletz, Data curation, Formal analysis, Investigation, Writing—original draft, Writing—review and editing; Alice Cezanne, Data curation, Investigation, Methodology, Writing—original draft, Writing—review and editing; Giambattista Guaitoli, Data curation, Investigation, Methodology, Writing—review and editing; Francesco Raimondi, Conceptualization, Data curation, Formal analysis, Investigation, Methodology, Project administration, Writing—review and editing; Marc Gentzel, Methodology, Project administration, Writing—review and editing; Vikram Alva, Supervision, Investigation, Methodology, Writing—review and editing; Michael Habeck, Investigation, Methodology, Writing—review and editing; Yannis Kalaidzidis, Data curation, Formal analysis, Methodology, Writing—review and

editing; Marius Ueffing, Supervision, Writing—review and editing; Andrei N Lupas, Supervision, Project administration, Writing—review and editing; Christian Johannes Gloeckner, Data curation, Formal analysis, Supervision, Writing—review and editing; Marino Zerial, Conceptualization, Data curation, Formal analysis, Supervision, Investigation, Methodology, Project administration, Writing—review and editing

### Author ORCIDs

Janelle Lauer (iD) https://orcid.org/0000-0003-1412-6766
Marc Gentzel (iD) https://orcid.org/0000-0002-4482-6010
Vikram Alva (iD) http://orcid.org/0000-0003-1188-473X
Andrei N Lupas (iD) http://orcid.org/0000-0002-1959-4836
Christian Johannes Gloeckner (iD) http://orcid.org/0000-0001-6494-6944
Marino Zerial (iD) https://orcid.org/0000-0002-7490-4235

### Decision letter and Author response

Decision letter https://doi.org/10.7554/eLife.46302.sa1
Author response https://doi.org/10.7554/eLife.46302.sa2

## Additional files

### Supplementary files

• Supplementary file 1. The results of the last round of modeling are shown comparing the top scoring clusters from each condition with the cross-linking-MS data.

• Supplementary file 2. Key resources table.

### Data availability

Data generated for Figures 1a and 3 are included in the supporting files.

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
