## [Decision Letter]

**Acceptance summary:**

Traffic through the endocytic pathway requires Rab5 GTPase, whose activation is regulated by the Rabex5 guanine exchange factor (GEF). Here the authors use hydrogen deuterium exchange mass spectrometry and chemical cross-linking mass spectrometry to correlate structural changes with nucleotide exchange activity. They show for the first time that ubiquitin binding serves not only to position Rabex5 on early endosomes but to also control its Rab5 GEF activity through allosteric structural alterations, both in vitro and in cells.

**Decision letter after peer review:**

Thank you for submitting your article "Allosteric structural alterations and auto-regulation of Rab5 GEF activity in Rabex5" for consideration by *eLife*. Your article has been reviewed by 3 peer reviewers, including Suzanne Pfeffer as the Reviewing Editor and Reviewer #1, and the evaluation has been overseen by Vivek Malhotra as the Senior Editor. The following individuals involved in review of your submission have agreed to reveal their identity: David Lambright (Reviewer #3).

The reviewers have discussed the reviews with one another and the Reviewing Editor has drafted this decision to help you prepare a revised submission.

This is a careful structural analysis of the full length Rab5 GEF, Rabex5 using mass spec with crosslinking and deuterium exchange mass spec to obtain clues to the interaction between Rab domains and Rabaptin5. The authors present a revised view of auto-inhibition and show for the first time that ubiquitinated cargoes may themselves regulate GEF activity rather than just serving as targeting elements. This latter finding will be of special interest to biologists interested in Rab5 events and should be made more prominent in the title and summary. Without this, the story is told in a very Rabex5 centric way that will be less broadly appreciated by the overall field of membrane trafficking. Indeed, additional cell biologically relevant discussion would be essential to make the story more broadly appreciated by the cell biology community overall.

We would like to publish this story in *eLife*, but feel that its impact would be significantly enhanced by the inclusion of any experiment in cells that would bolster the idea that ubiquitin recognition alone enhances GEF activity. Can you please include any experimental data that would support this conclusion enough to change the title of the story for the broadest readership? Testing these ideas in cells, using mutants that uncouple ubiquitin-dependent membrane recruitment from allosteric activation of the GEF, would add a great deal in terms of the overall significance of the work. We are excited about the story and are confident you can add this without too much difficulty.

Other comments are for the most part textual and follow here.

1) The Introduction discusses UBD and MIU but the figure shows one bar. Please include both domain names in the text above the red bar so reader can follow more easily. Please name domains on Figure 1, above colored bars. Please make clear that both MIU and UBD are in the same red bar.

2) Please show Figure 1A model in relation to current domain PDB structures used to derive structure model so reader can discern differences. Figure 1B please indicate N- and C-termini.

3) Figure 4 is hard for a non-expert to follow. What differences are being highlighted? What quantity is represented by the scale bar in Figure 4 and Figure 6? Where should the reader look? Figure 5C. Fraction of control *rate*? Please use a clearer Y axis label. Figure 5A, B; please include the name of the construct on the figure with tetraUb concentration indicated for easy comparison with C.

4) Parts of the text are repetitive and need to be rewritten for clarity and to eliminate jargon. Some referencing is incomplete or not inserted. We listed some issues but there are more.

4.1) Abstract: "Here, we studied full-length Rabex5 and Rabaptin5 proteins as well as domain deletion Rabex5 mutants using hydrogen deuterium exchange mass spectrometry. We generated a structural model of Rabex5, using chemical crosslinking mass spectrometry and integrative modeling techniques. Our results are inconsistent with the previous model of auto-inhibition."

Methods are introduced but the key findings and conclusions drawn from the use of the method are not explained. The emphasis that previous models for regulation are not consistent with the data is not particularly insightful. Focussing on what the new conclusions are would be better.

4.2) For example: "Utilizing a structural proteomics approach, we used integrative modeling techniques to combine chemical crosslinking mass spectrometry (XL-MS) and available x-ray crystallographic structures to generate a structural model of full-length Rabex5 (Figure 1)."

Did the authors use structural proteomics or integrative modelling? It isn't clear from the text. In fact, they used integrative modelling to combine structural proteomics data with the available crystal structures of domains of Rabex5. In this case the structural proteomics data was cross-link mass spectrometry & later hydrogen-deuterium exchange).

Write out cross-link/cross-linking in full, rather than abbreviating to "XL", which often has other meanings.

4.3) Introduction. "In addition, Rabex5 forms a relatively stable complex with Rabaptin5 (REF Horiuchi), raising the question of whether this allosteric mechanism is the key modulator of the GEF activity."

The reference to Horiuchi is not inserted.

4.4) The Introduction references G-protein regulation in a very general sense, rather than focussing on Rabs and their regulators. The recent review from Müller and Goody (2018) covers the role of GDI and GEFs in Rab regulation, including many structural details of the mechanisms.

4.5) Introduction. "Our results led us to propose a novel allosteric mechanism for regulating the nucleotide exchange catalytic process."

The tense is incorrect, "our results lead us to propose….". Allosteric mechanisms are not "novel", so this claim should be removed. The details for any given example may be different, but that is a different matter.

5) Figure 3B and Materials and methods section. It appears that k_cat_/K_m_ is calculated from kinetic data obtained for a single enzyme concentration, so in Figure 3B what is plotted is in fact k_obs_/[enzyme concentration]. This is more normally done using enzyme titration. Observed rate constants are plotted as a function of enzyme concentration, the slope is k_cat_/K_m_ and the basal rate is the intersect on the Y-axis.

6) One technical complication is that the reaction starts when Rabex5 is added, and if different proteins have higher or lower GEF activity then by the time of the first measurement the values are already different. This complicates fitting an initial rate to the data, since the top part of the curve is lost. In Figure 3A, Rab has different amounts of bound nucleotide for the different conditions at t0 suggesting this issue is present in the data.

7) The authors write that they model full-length Rabex5. However, the model in Figure 1B appears to lack the first 20aa and last 42aa if the colour-coded schematic in Figure 1A is correct. Are the white segments in the model in Figure 1B? If the model isn't full-length, then remove the claim from the text.

8) Figure 5, plot rate as a function of Ub-concentration. It would also be helpful to show if this effect is unique to K63 chains and what effect mono-ubiquitin has.

9) Exploratory HDX-MS results to select RabexΔ82-117 for use is a bit vague. Can the authors be more specific about why this region was selected? Apparently, a larger region was slightly destabilizing with respect to deuterium uptake, but it isn't clear if shorter deletions were tested and what the results were.

10) Would the differences in the orientation of the helical bundle relative to the catalytic core be expected to directly impact the accessibility of the catalytic site to Rab5 or is it more likely that the differences in GEF activity are a consequence of conformational changes in the substrate site of the Vps9 domain?

11) Indicate the type of residue being cross-linked (e.g. cysteine).

---

## [Author Response]

We would like to publish this story in eLife, but feel that its impact would be significantly enhanced by the inclusion of any experiment in cells that would bolster the idea that ubiquitin recognition alone enhances GEF activity. Can you please include any experimental data that would support this conclusion enough to change the title of the story for the broadest readership? Testing these ideas in cells, using mutants that uncouple ubiquitin-dependent membrane recruitment from allosteric activation of the GEF, would add a great deal in terms of the overall significance of the work. We are excited about the story and are confident you can add this without too much difficulty.

We have addressed this request by generating a Rabex5 KO HeLa cell line and expressing full length Rabex5, the RabexΔUBD mutant and the RabexΔLinker mutant as control. The results shown in a new Figure 8 nicely show that loss of Rabex5 causes an alteration of the early endosomal network that can be rescued by low level re-expression of Rabex5. At high levels of expression, Rabex5 causes a gain of function (GOF) phenotype as appearance of enlarged early endosomes. The RabexΔUBD mutant also rescued the Rabex5 KO phenotype at low expression levels and yielded a GOF phenotype at high levels, despite the fact that it was less efficiently recruited to early endosomes than Rabex5 (Figure 8C). This is consistent with the high constitutive GEF activity and lack of UBD contributing to the endosomal localization of this mutant.

In contrast, the RabexΔLinker rescued the Rabex5 KO phenotype, but was much less potent on the stimulation of early endosome size, despite its localization to early endosomes (Figure 8C), consistent with its lower catalytic activity. Altogether, these results indicate that the Rabex5 UBD regulates the association of Rabex5 to early endosomes and its GEF catalytic activity, as its removal leads to a high-level activity despite a low association with early endosomes.

Other comments are for the most part textual and follow here.1) The Introduction discusses UBD and MIU but the figure shows one bar. Please include both domain names in the text above the red bar so reader can follow more easily. Please name domains on Figure 1, above colored bars. Please make clear that both MIU and UBD are in the same red bar.

Figure 1 has been updated to include both UBD and MIU names. Subsequent figures do not include the distinction between UBD and MIU because throughout the manuscript they are treated as one unit.

2) Please show Figure 1A model in relation to current domain PDB structures used to derive structure model so reader can discern differences. Figure 1B please indicate N- and C-termini.

N- and C-terminal labels were added to Figure 1 and Figure 1—figure supplement 1 was created to illustrate the superimposition of our model on 1TXU.

3) Figure 4 is hard for a non-expert to follow. What differences are being highlighted? What quantity is represented by the scale bar in Figure 4 and Figure 6? Where should the reader look? Figure 5C. Fraction of control rate? Please use a clearer Y axis label. Figure 5A, B; please include the name of the construct on the figure with tetraUb concentration indicated for easy comparison with C.

Multiple changes were made to Figure 4, Figure 5, and Figure 6 to address the points made here. Another panel was added to Figure 4 and to Figure 6 to line up regions within Rabex5 and how they are altered in the different mutants. This makes it clearer for non-experts of Rabex5 structure. Also, information was added in the text to help illustrate which panels in Figure 4 are relevant for each sentence.

4) Parts of the text are repetitive and need to be rewritten for clarity and to eliminate jargon. Some referencing is incomplete or not inserted. We listed some issues but there are more.[…]

In short, numerous sections of text were removed to address repetition. The references were updated with the Goody review and the Horiuchi error was corrected.

5) Figure 3B and Materials and methods section. It appears that k_cat_/K_m_ is calculated from kinetic data obtained for a single enzyme concentration, so in Figure 3B what is plotted is in fact k_obs_/[enzyme concentration]. This is more normally done using enzyme titration. Observed rate constants are plotted as a function of enzyme concentration, the slope is k_cat_/K_m_ and the basal rate is the intersect on the Y-axis.

There is some confusion regarding the accuracy of the axis label in Figure 3. Kinetic data were obtained from a single enzyme concentration. In accordance with previous work in the literature, we labeled these results as k_cat_/K_m_. However, a reviewer has suggested that this is more accurately referred to as k_obs_/[enzyme]. The labels have been updated accordingly.

6) One technical complication is that the reaction starts when Rabex5 is added, and if different proteins have higher or lower GEF activity then by the time of the first measurement the values are already different. This complicates fitting an initial rate to the data, since the top part of the curve is lost. In Figure 3A, Rab has different amounts of bound nucleotide for the different conditions at t0 suggesting this issue is present in the data.

We are aware of this limitation and our analyses include a non-linear curve fit of the data. Thus, the k_obs_ values do not rely solely on the linear portion of the curve. As such, our k_obs_ values for the most active enzymes may be slightly underestimated and we state that their activities are “at least x-fold higher” to account for these minor discrepancies. In no way does this alter any of our conclusions.

7) The authors write that they model full-length Rabex5. However, the model in Figure 1B appears to lack the first 20aa and last 42aa if the colour-coded schematic in Figure 1A is correct. Are the white segments in the model in Figure 1B? If the model isn't full-length, then remove the claim from the text.

The coloring schemes used in Figure 1A and 1B were confusing and has been altered to be clearer. The Rabex5 in Figure 1B is indeed a full-length model and we thank you for pointing out this problem.

8) Figure 5, plot rate as a function of Ub-concentration. It would also be helpful to show if this effect is unique to K63 chains and what effect mono-ubiquitin has.

Another panel was added to Figure 5 to show the non-corrected data (Figure 5D). Due to the activity differences between the WT Complex and RabexΔUBD Complex, we chose to normalize the activities. The normalized data is still included, but may be redundant. We also tested Lys64 linked tetra-Ubiquitin as well as a linear construct of di-Ubiquitin. A description of those results has been added to the text. The linear construct did not alter Rabex5 activity. Thus, mono-Ubiquitin would likely have no activity.

9) Exploratory HDX-MS results to select RabexΔ82-117 for use is a bit vague. Can the authors be more specific about why this region was selected? Apparently, a larger region was slightly destabilizing with respect to deuterium uptake, but it isn't clear if shorter deletions were tested and what the results were.

We added text to help describe our thought processes behind making the Linker deletion mutants. Only two deletion constructs were made because the exploratory HDX-MS results which guided the design were unable to isolate the activity to a smaller region within the linker. We didn’t want to embark on constructing mutants without a rational reason.

10) Would the differences in the orientation of the helical bundle relative to the catalytic core be expected to directly impact the accessibility of the catalytic site to Rab5 or is it more likely that the differences in GEF activity are a consequence of conformational changes in the substrate site of the Vps9 domain?

We have added a sentence stating that we believe this not the case as the 4-HB is likely on the opposite side of the protein. Thus, alterations in activity are likely due to conformational changes in the Vps9 domain rather than some sort of competition.

11) Indicate the type of residue being cross-linked (e.g. cysteine).

The name of our cross-linking reagent is listed in the Materials and methods section. It reacts with Lys.